# Precision Science on Incidence and Progression of Early-Detected Small Breast Invasive Cancers by Mammographic Features

**DOI:** 10.3390/cancers12071855

**Published:** 2020-07-10

**Authors:** Rene Wei-Jung Chang, Shu-Lin Chuang, Chen-Yang Hsu, Amy Ming-Fang Yen, Wendy Yi-Ying Wu, Sam Li-Sheng Chen, Jean Ching-Yuan Fann, Laszlo Tabar, Robert A. Smith, Stephen W. Duffy, Sherry Yueh-Hsia Chiu, Hsiu-Hsi Chen

**Affiliations:** 1Institute of Epidemiology and Preventive Medicine, College of Public Health, National Taiwan University, Taipei City 100, Taiwan; d06849009@ntu.edu.tw (R.W.-J.C.); bacilli65@gmail.com (C.-Y.H.); 2Department of Medical Research, National Taiwan University Hospital, Taipei City 100, Taiwan; d99849010@ntu.edu.tw; 3School of Oral Hygiene, College of Oral Medicine, Taipei Medical University, Taipei City 110, Taiwan; amyyen@tmu.edu.tw (A.M.-F.Y.); samchen@tmu.edu.tw (S.L.-S.C.); 4Department of Radiation Sciences, Oncology, Umeå University, 90187 Umeå, Sweden; wendy.wu@umu.se; 5Department of Health Industry Management, College of Healthcare Management, Kainan University, Taoyuan City 338, Taiwan; jeanfann@mail.knu.edu.tw; 6Department of Mammography, Falun Central Hospital, 791823 Falun, Sweden; laszlo@mammographyed.com; 7Center for Cancer Screening, American Cancer Society, Atlanta, GA 30303, USA; robert.smith@cancer.org; 8Centre for Cancer Prevention, Queen Mary University of London, Charterhouse Square, London EC1M 6BQ, UK; s.w.duffy@qmul.ac.uk; 9Department of Health Care Management, College of Management, Chang Gung University, Taoyuan City 333, Taiwan; 10Division of Hepatogastroenterology, Department of Internal Medicine, Kaohsiung Chang Gung Memorial Hospital, Kaohsiung City 833, Taiwan

**Keywords:** breast cancer, interval cancer, mammography, sensitivity, sojourn time

## Abstract

The aim was to evaluate how the inter-screening interval affected the performance of screening by mammographic appearances. This was a Swedish retrospective screening cohort study with information on screening history and mammography features in two periods (1977–1985 and 1996–2010). The pre-clinical incidence and the mean sojourn time (MST) for small breast cancer allowing for sensitivity by mammographic appearances were estimated. The percentage of interval cancer against background incidence (I/E ratio) was used to assess the performance of mammography screening by different inter-screening intervals. The sensitivity-adjusted MSTs (in years) were heterogeneous with mammographic features, being longer for powdery and crushed stone-like calcifications (4.26, (95% CI, 3.50–5.26)) and stellate masses (3.76, (95% CI, 3.15–4.53)) but shorter for circular masses (2.65, (95% CI, 2.06–3.55)) in 1996–2010. The similar trends, albeit longer MSTs, were also noted in 1977–1985. The I/E ratios for the stellate type were 23% and 32% for biennial and triennial screening, respectively. The corresponding figures were 32% and 43% for the circular type and 21% and 29% for powdery and crushed stone-like calcifications, respectively. Mammography-featured progressions of small invasive breast cancer provides a new insight into personalized quality assurance, surveillance, treatment and therapy of early-detected breast cancer.

## 1. Introduction

Breast cancer can be detected in the pre-clinical detectable phase (PCDP) through an appropriate screening modality such as mammography so as to lead to mortality reduction from breast cancer by 13–33% [1,2,3,4].

A number of previous studies have elucidated the temporal natural history of breast cancer, free of breast cancer, PCDP, clinical phase (CP) [5,6,7,8] and the extended process with the incorporation of tumor attributes such as node status [9,10,11] by modeling the average duration between PCDP and CP, named as the mean sojourn time (MST) [6,12,13]. However, little is known about the temporal natural history stratified by mammographic features that have been regarded as an independent predictor for the prognosis of breast cancer [14,15]. Moreover, due to the widespread use of mammography screening for breast cancer, elucidating the natural history of breast cancer by mammographic features with emphasis on breast tumors less than 1–14 mm is of paramount importance [16,17,18]. The recently proposed classification of breast cancer by the anatomic site of cancer origin also revealed the two types of breast cancer, acinar adenocarcinoma of the breast (AAB) and ductal adenocarcinoma of the breast (DAB) [19]. The origin of the former is derived from TDLU of acinar whereas the origin of the latter is derived from the duct of breast. The AAB type, including the stellate type, the circular type, the powdery type, and the crushed stone-like type, can be prevented by early detection through mammography, but the latter had poor prognosis and may not be amenable to early detection through screening [19,20]. Among AAB, the stellate type, the powdery type, and the crushed stone-like had better prognosis than the circular type [19].

It is therefore interesting to estimate mammography-specific MST for breast tumors less than 1–14 mm, focusing on AAB only, before and after widespread use of mammography based on a large population-based cohort data. These estimates of MST together with pre-clinical incidence rate of cancer, making allowance for sensitivity, enable one to estimate the expected (background) incidence (E) in the absence of screening in order to quantify the I/E ratio defined as the observed interval cancer (I) as a percentage of the expected (E) given the inter-screening interval and the detectability of mammography (sensitivity). The effect of the inter-screening interval and sensitivity on the I/E ratio was simulated by various scenarios of two quantities in order to provide precision surveillance for early detection of a small breast tumor.

## 2. Results

### 2.1. Natural History of Breast Cancers by Mammographic Features

The percentage of breast cancer cases by different mammographic features and tumor size for each of the two periods is shown in Figure 1. The percentage of small breast tumor (<15 mm) in the second period (1996–2010) was almost double of that in the latter period. Among these small tumors, the percentage of powdery, crushed stone-like, casting and architectural distortion in the second period was almost double compared to that in the first period.

The estimates of preclinical incidence rates, MST and sensitivity by mammographic appearances for breast tumors of size smaller than 15 mm in diameter and for large tumors are shown in Figure 2. The pre-clinical incidence rate (per 100,000) from PCDP to CP was the highest for the stellate masses, followed by circular masses, and powdery and crushed stone-like calcifications in the first period. It is also very interesting to note that the pre-clinical incidence rate for a small breast tumor in total increased from the first period to second period by 26% whereas that for a large breast tumor was reduced by 44%.

The MST was longest for the stellate masses, followed by the powdery and crushed stone-like calcifications and the circular masses. The shorter MST for breast tumors larger than 15 mm was seen. In the period between 1977 and 1985, the estimate of sensitivity was lower for stellate masses compared with other types of small breast tumor, but it was even lower for breast tumors of size 15 mm or larger as the cutoff of a large tumor here is not based on 2 cm or larger as defined in the conventional manner.

Figure 2 also shows the estimates of MST between 1996 and 2010 were shorter than those between 1977 and 1985 because the time horizon for CP was advanced due to the high awareness of women diagnosed as interval cancer or non-participants. However, the sensitivity was substantially improved in the second period.

Figure 3a shows cumulative risk curves for the transition from PCDP to CP in the absence of screening, for the three mammographic appearance types and for breast tumors larger than 15 mm estimated from the first period. The risk of progressing to symptoms was the highest for tumors of size 15 mm or more. The circular masses had more rapid progression than two other types, the stellate masses and the powdery and crushed stone-like calcifications.

Five-year cumulative risks of progression to symptomatic disease without early detection were 0.83 for stellate masses, 0.94 for circular masses, 0.85 for powdery and crushed stone-like calcifications and 0.97 for tumor sizes of 15 mm or larger. The median time to CP was 1.69 years for stellate masses, 1.33 years for circular masses, and 1.63 years for powdery and crushed stone-like calcifications.

Figure 3b shows the cumulative risk curves for the transition from PCDP to symptomatic cancer for size < 15 mm by mammographic features, and for size 15 mm or larger in the second period. The estimates are similar to the figures for the first period but the risk of progression for the circular type lower than that for the breast tumors of size 15 mm or larger were remarkably noted in the second period between 1996 and 2010.

### 2.2. Empirical Estimates of I/E Ratio by Mammographic Features

Using the parameters estimated from Figure 2 in the second period (1996–2010), Figure 3a (see the indicated points by arrows; the details of numerical values also see the bottom panel of Table 1) shows the percentages of I/E ratio for the stellate type were 23% and 32% for the biennial and triennial program, respectively. The corresponding figures were 32% and 43% for the circular type and 21% and 29% for powdery and crushed stone-like calcifications. In spite of longer MSTs, the similar findings on the percentage of the I/E ratio were also noted in the first period (Figure 4b).

### 2.3. Interval Cancer of Different Mammographic Features by Inter-Screening Intervals and Sensitivity

Table 1 shows the simulated results for interval cancer incidence (I) as a percentage of the expected incidence from the control group (E) and the estimates of (I/E) × 100% by mammographic appearances in comparison with breast tumors larger than 15 mm in the two periods respectively. The longer the inter-screening interval the higher the I/E ratio was. The higher the sensitivity, the lower the I/E ratio (see Figure 4).

The I/E ratios for the stellate masses and the powdery and crushed stone-like calcifications were smaller compared with breast tumors of size 15 mm or larger, whereas the corresponding I/E ratios for the circular type were similar to those for the largest tumors (larger than 15 mm). Note that the shorter the inter-screening interval and the higher the sensitivity, the larger the reduction in interval cancers. In the stellate masses, even when sensitivity was only 40%, a 46% reduction in proportional interval cancer rates was observed for a triennial program and 76% for an annual program. A larger reduction of 92% for an annual program and 79% for a triennial program was estimated when sensitivity was 100%. Similar findings with a lesser reduction were found for powdery and crushed stone-like calcifications.

The I/E ratio was higher in the second than in the first period, as the mean sojourn time was shorter in the second period. However, the trends in the second period regarding the association between inter-screening interval and sensitivity were the same as in the first period. Compared with breast tumors of size 15 mm or larger, the reduction in the I/E ratio was the greatest for powdery and crushed stone-like calcifications, and slightly lower for stellate masses. Both types showed larger reductions than the circular masses. Note that the reduction in the I/E ratio for stellate masses in the second period was smaller than that in the early period. Unlike the results in 1977–1985, the I/E ratios for the circular type were smaller than those for breast tumors of size 15 mm or larger.

## 3. Discussion

This is the first study to quantify the temporal natural history of small breast cancers (1–14 mm) by mammographic appearances using longitudinal follow-up data. These estimates of MST not only grasped a better understanding of the heterogeneity of the biological properties of breast tumors but also provided a new insight into precision screening policy and medical surveillance for breast tumors with different appearances on mammographs.

Based on the estimates of MSTs, it reveals the heterogeneity of breast tumors by mammographic appearance in biological growth rate. Breast tumors featured as stellate masses and powdery and the crushed stone-like calcifications on the film of mammography have slow progressions compared to circular masses. This is supported by the breast tumor growth displayed by the chronological order of morphological findings on the mammogram. Figure 3 shows a radiological illustration of tumor growth displayed on the mammograms identified, retrospectively, for the three types, showing slower progression to symptomatic diagnosis.

Compared with the estimates of MST in the trial period (1977–1985), the corresponding estimates in the service screening period (1996–2010) were similar for the powdery and crushed stone-like calcification but shorter for stellate masses and for circular masses after adjustment for mammography-featured sensitivity. These shorter sojourn times are not a consequence of more rapid progression in the second period, but a reflection of greater awareness of women to seek medical care. In the second period, it is also interesting to note that the estimated sensitivity was improved to 95% or above regardless of mammographic appearance.

Information on the MST for each type of mammographic appearance can be used to predict the risk of transition to symptomatic disease in the absence of early detection measures. For small breast cancers, the circular masses had the most rapid progression and closer to that of breast cancers of size 15 mm or larger (see Figure 3a). Such a finding provides information on medical consultation when women were detected with breast cancer with different types of mammographic appearances. In addition, knowing a specific type of mammographic appearance also provides guidance for the future surveillance of women diagnosed with breast cancer, particularly small breast cancer [16,21,22]. If the tumor appears as a circular mass, a more intensive surveillance schedule is required, and further biomarker tests might be required, with a view to providing a further individually tailored therapy. In contrast, small breast cancers appearing as stellate masses or powdery and the crushed stone-like calcifications could be less aggressively managed, in order to avoid overtreatment.

The findings specific to their mammographic features have the potential to inform decisions on screening, treatment, and the follow-up of cancers. Since we cannot predict who is likely to develop which mammographic type of cancer, it would seem prudent to design screening programs for the most rapidly progressing circular masses. However, an individually-tailored inter-screening policy can be made if a further study on the identification of genetic variants [23,24,25] and risk factors in association with different types of mammographic features can be implemented. Treatment and surveillance of those with diagnosed cancer, however, can be stratified according to the threat posed by the tumor, as determined by mammographic appearance in addition to pathological findings.

Our study used Tabar’s classification of mammographic features, which categorized malignant tumors, like the Breast Imaging Reporting and Data System (BI-RADS) [26] categories 4 (suspicious abnormality) and 5 (highly suggestive of malignancy) and later with biopsy proven malignancy, according to the morphology of mammography findings. This classification has also been correlated with the BI-RADS descriptor as delineated in the section of data collection.

### Precision Screening Policy on Inter-Screening Interval and Sensitivity

The application of transition parameters and sensitivity from a three-state Markov model to the simulation of different inter-screening intervals, and also the ability to detect early breast cancer (sensitivity), provides a new insight into how to develop a precision screening policy. The importance of early detection and treatment of the more rapidly progressing circular type would suggest a shorter interval (since we cannot tell who will develop cancer by mammographic appearance). This in turn would maintain a good outcome in terms of low interval cancer rates for stellate masses which had lower sensitivity in the early period (Figure 2). Our simulation results based on the transition parameters and sensitivity from the three-state Markov model give a clue to designing different combinations of inter-screening intervals and sensitivity in order to optimize the benefits given limited resources. This can be illustrated as follows.

If the criterion for proportional interval cancer incidence is to be less than 40%, then this could be achieved with a two-year inter-screening interval given at least 80% sensitivity is anticipated. If the criterion for the proportional interval cancer incidence is to be less than 30%, to achieve this for all mammographic types would require near 100% sensitivity and a two-year inter-screening interval. If 60–80% sensitivity is anticipated, annual screening would be required.

There are three limitations. First, as noted above, we cannot tell who will develop which type of cancer, so we cannot stratify screening regimens to mammographic appearance. Nonetheless, our finding provides a new insight into the development precision preventive model for the identification of individual characteristics in relation to each own mammographic feature. Our model cannot accommodate small breast cancers displaying several types by mammographic appearance e.g., a stellate synchronous with a circular. The models presented here were stratified by tumor size. For the second period, however, we were able to model simultaneous progression by size as well as symptomatic status. This will be the subject of a separate communication. Finally, our mammographic-feature-specific natural history model has not included ductal adenocarcinoma of the breast (DAB) and the casting-type calcifications and architectural distortions which are characteristic of DAB. How to incorporate DAB to complete the estimation of mammographic feature-specific breast cancer natural history is the subject of ongoing research.

Third, modern histopathological types as well as molecular subtypes of tumors have been proven to be associated with different mammographic features. Our previous study has already demonstrated their roles in initiators of breast cancer and promoters to cancer progression [27,28], it is of paramount importance to incorporate such information into multi-state disease natural history to enrich precision science on the prevention of breast cancer. However, the current study is meant to compare the disease natural history of small breast tumors by mammographic appearances between the trial period and the service period. Two reasons preclude us from adding such information to the disease natural history of breast cancer. First, there is a lack of information on the emerging molecular classification in the first trial period (1977–1985). The comparison of these phenotypes between two periods would not be possible. Second, as we focus on the disease natural history of small breast cancers rather than all breast cancers, large samples are, therefore, required if these new phenotypes are further considered. In a similar manner, the current study is also limited to allow for the incorporation of new mammographic techniques and alternative imaging techniques, which have also been tested as screening tools, such as tomosynthesis [29,30,31,32,33], contrast enhanced spectral mammography (CESM) [34,35], magnetic resonance imaging (MRI) [36,37], ultrasound [37,38] and also fully automated support systems and AI systems [39,40]. In spite of the lack of such information, we believe the expedient use of fully automated systems such as computer-aided systems may yield sensitivity as favorable as that operated by the top radiologist (Tabar Laszlo), one of the co-authors, in the second period when such a fully automated system is still immature. This is particularly true for the incorporation of artificial intelligence (AI) with machine learning [34,40] to provide a grand breakthrough. Hence, doing so may alleviate the burden of manpower on well-trained radiologists when a mass service screening program is involved. Our proposed novel model on multi-state disease natural history by mammographic features is easily adapted to incorporate such new information to elucidate their roles in precision prevention of breast cancer in the future.

## 4. Materials and Methods

### 4.1. Study Framework and Design

The conceptual study framework used here is diagrammed in Figure 5 to elucidate how mammographic appearances (main independent variables), one of imaging phenotypes of breast tumors, are related to three main outcomes (the bottom panel of Figure 5): the pre-clinical incidence rate of cancer, the progression rate from PCDP to CP and the sensitivity by mammographic appearances for 1–14 mm tumors. The former two outcomes represent the natural disease progression of breast cancer in the absence of screening. The third outcome of sensitivity is related to the detectability of mammography. The estimates of the three outcomes rely on information on detection modes (the right column of Figure 5). Three outcomes were also considered for the progression of large tumors (15 mm or larger) without being stratified by tumor appearances because once a tumor becomes larger, mammographic appearances are not of great interest in our study. The estimates of the progression rate from PCDP to CP and sensitivity yield the sensitivity-adjusted MST of mammography by mammographic appearance and tumor size. Pre-clinical incidence and sensitivity-adjusted MSTs were used to compute the expected incidence rate in the absence of screening. This is the denominator of interval cancer as a percentage of the expected incidence rate (I/E) ratio. This parameter is the main indicator for the evaluation of the effect of the inter-screening interval and sensitivity.

We further used a simulated randomized controlled design to investigate the impact of the inter-screening interval and sensitivity given a study population on women aged 40 years and older randomly allocated to four arms, annual regime, biennial regime, triennial regime and no screening. Each screening regime was simulated to have heterogeneous ability to detect breast cancer depending on the mammographic appearance, tumor size, use of screening modalities and operator’s performance. The percentage of the I/E ratio was adopted as an outcome to assess the effect of the inter-screening interval in combination with the sensitivity on the performance of the screening.

### 4.2. Study Subjects

Study subjects were derived from the retrospective cohort pertaining to breast cancer screening in Dalarna County, Sweden in two periods: the first period in the era of trial (1977–1985) and the second period (1996–2010) in the era of service. Study subjects in the early period were composed of women who were invited to a population-based randomized controlled trial on breast cancer screening with mammography in Dalarna (formerly called Kopparberg), one of the Swedish Two-County Trial between 1977 and 1985. The details of the design of this trial (called the Dalarna trial hereafter) have been described in full elsewhere [41]. In brief, women aged 40–74 years or older were randomized to two arms, the invited and the uninvited with and without single-view mammography. The inter-screening interval was 24-months for those aged 40–49 years and 33 months for women aged 50 years or more. Mammograms were retrieved and mammographic appearances were classified for 251 cancers of size less than 15 mm and 384 cancers of size larger than 15 mm.

After the trial period, the nationwide mammography screening for breast cancer in Sweden with two-view mammography and an inter-screening interval ranging from 18 months to 3 years for women aged 40 years and older was launched in Sweden. However, as mammography service screening had been offered after 1986, information on mammographic appearance had been recorded in medical charts but not in electronic form until 1996. Information on mammographic appearance on 2204 tumors of all sizes from 1996 to 2010 was, therefore, available for the following analysis. This study was approved by the Ethical Committee of Taipei Medical University (TMU-JIRB No.: N201607008).

### 4.3. Data Collection

To estimate the parameters of interest, we collected information including the date of diagnosis, detection mode and mammographic appearance. The mammographic appearance was classified in light of Tabar’s categories [14,42] as shown in Figure 6, including stellate, circular, powdery, crushed stone-like, casting and architecture distortion. As a matter of fact, the Tabar classification and the widely used BI-RADS system are commensurate. However, as the former has been long used in Falun Central Hospital, Sweden, since 1977 where data from both the trial period and the service period were collected, we retained the language of Tabar’s classification throughout the text. Nonetheless, the corresponding morphologic descriptors for mass, margin, calcifications and distribution between two classifications have been provided here without the loss of generality. A speculate tumor mass without associated calcifications was classified as a stellate mass (BI-RADS: irregular shaped mass with speculated margin). A round or oval-shaped mass was classified as circular (BI-RADS: oval or round shaped mass with circumscribed margin). On the other hand, a tumor mass with associated amorphous, non-casting type calcifications was classified as “powdery”, and a mass with granular-shaped calcifications was classified as “crushed stone-like” type (BI-RADS: pleomorphic-heterogeneous-granular-type or amorphous-indistinct-type calcifications). Other appearances such as casting calcifications (BI-RADS: fine linear branching or casting calcifications) and architecture distortion (the same as BI-RADS) are characteristic of DAB, both of which are not our interest in this study.

Interval cancers are composed of two types: newly diagnosed cancers from the time since the last negative screen and false negative cases missed at the last screen. Therefore, interval cancers together with screen-detected cancers are informative not only on the transition from PCDP to symptomatic disease but also on the sensitivity of mammography as shown in Figure 6.

### 4.4. Statistical Analysis

We applied a three-state Markov exponential regression model that has been used previously [4,5,9,10,12,21,43,44] to incorporate mammographic appearance as a covariate with two proportional hazards regression forms to estimate the pre-clinical incidence rate and the transition rate from the PCDP to symptomatic breast cancer by mammographic appearance. Our primary interest was in the natural history of breast tumors smaller than 1–14 mm by mammographic appearance; therefore, the transition rates for breast tumors larger than 15 mm were estimated without being stratified by mammographic appearance. The estimates of sensitivity by each mammographic appearance for 1–14 breast tumors and for all breast tumors larger than 15 mm were estimated. For the likelihood functions for estimating the transition parameters and their 95% confidence intervals, we applied the Bayesian Monte Carlo Markov Chain (MCMC) to producing posterior distributions for estimating the transition parameters and sensitivity using the WinBUGs program [45,46].

## 5. Conclusions

Elucidating the natural history of small breast tumors in the two periods revealed a decreasing trend of incidence on larger tumors but an increasing trend of incidence on small breast tumors and also shorter sojourn times in the second period that are a reflection of greater awareness of women. Among breast tumors by mammographic features, circular masses were the most rapid in progression and powdery and crushed stone-like were the slowest in progression. These findings not only reveal the heterogeneity of biological breast tumor growth but also provide a new insight into precision screening policy and medical surveillance for these early-detected small breast cancers.

## Figures and Tables

**Figure 1 cancers-12-01855-f001:**
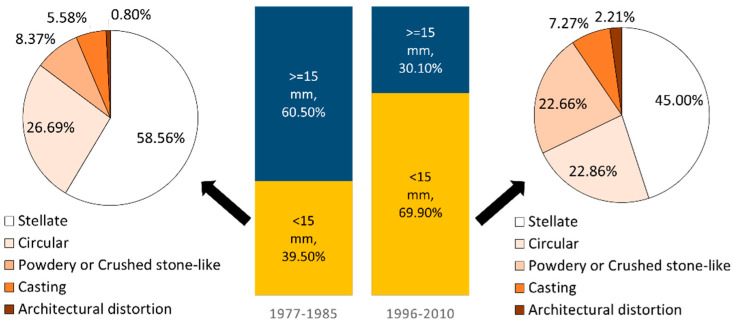
Percentage of breast cancer cases by different mammographic features and tumor size in 1977–1985 (*n* = 635) and 1996–2010 (*n* = 2204). The bar chart represents the percentage of different tumor size and the pie charts show the distribution of mammographic features among tumors smaller than 15 mm.

**Figure 2 cancers-12-01855-f002:**
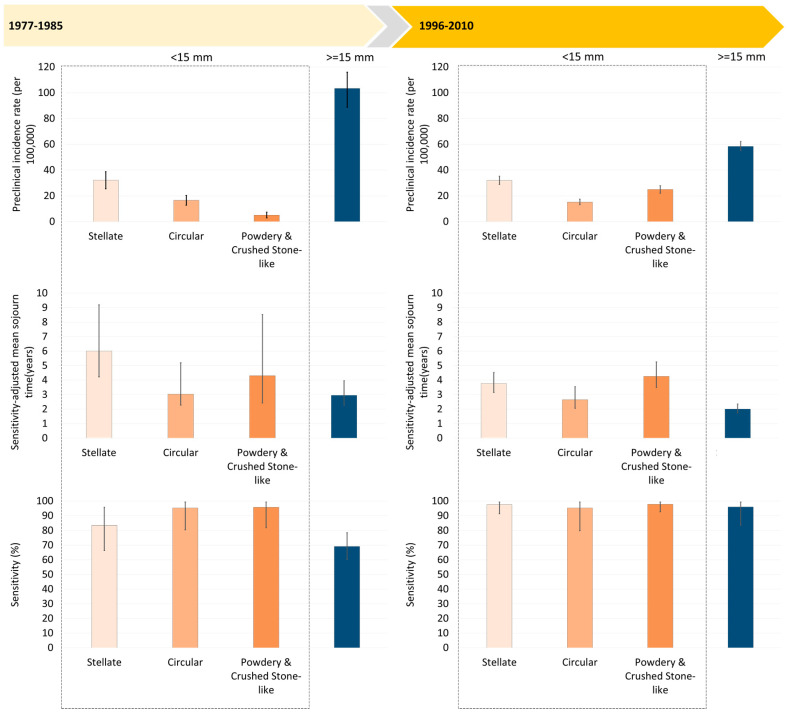
Mammography-featured natural history of breast cancers smaller than 15 mm and breast cancers larger than 15 mm.

**Figure 3 cancers-12-01855-f003:**
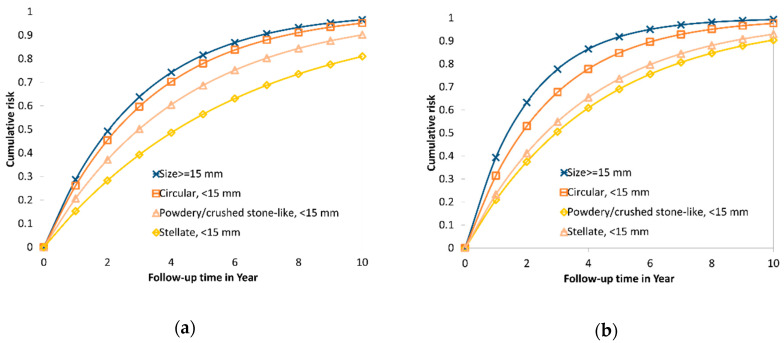
Cumulative risk from PCDP to CP without early detection for three types of mammographic appearance with tumor size <15 mm and ≥15 mm among AAB breast tumor in the two periods. (**a**) 1977–1985; (**b**) 1996–2010.

**Figure 4 cancers-12-01855-f004:**
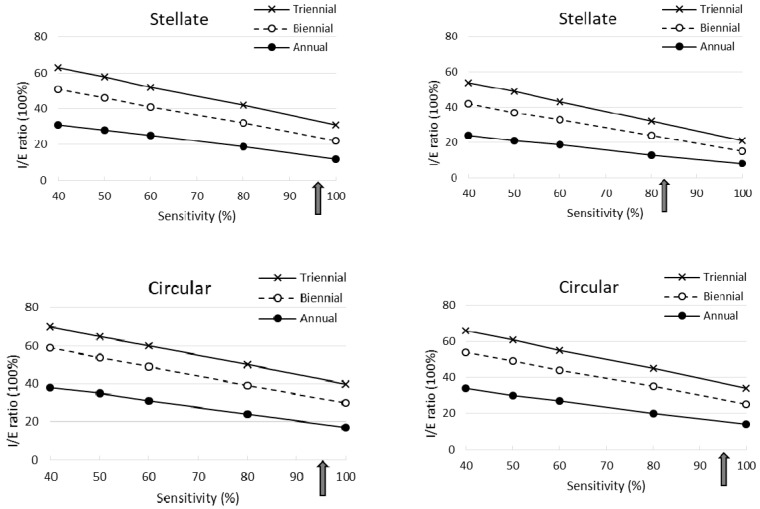
Simulated results with respect to proportional interval cancer incidence for breast cancer between 1977 and 1985 varying with inter-screening interval and sensitivity (arrows indicated the estimated sensitivity in this study). (**a**) 1996–2010; (**b**) 1977–1985.

**Figure 5 cancers-12-01855-f005:**
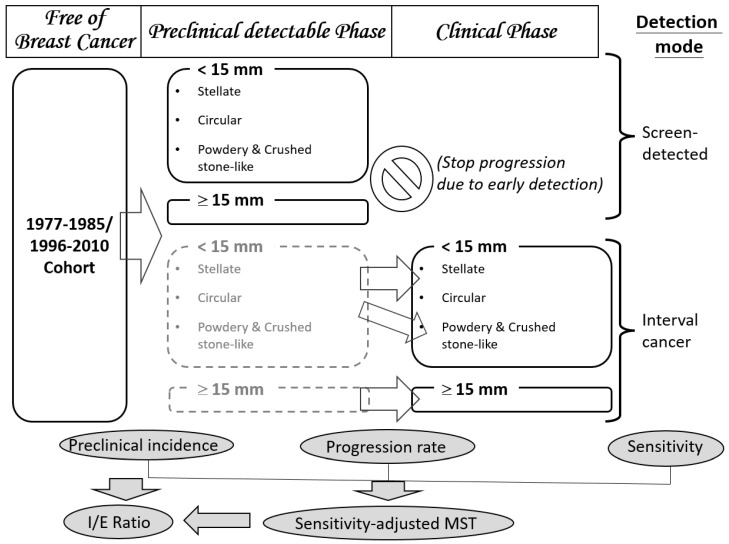
The study design relating disease status, mammographic appearances (main independent variables) and the three main outcomes of interests. (The box in gray dash boarder referred to unobserved status).

**Figure 6 cancers-12-01855-f006:**
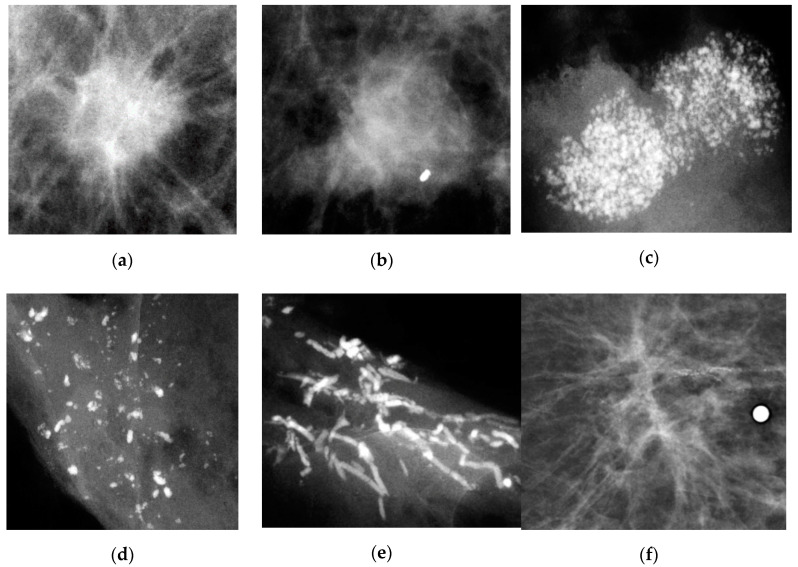
Types of mammographic appearance including Stellate, Circular, Powdery, Casting, Crushed stone-like and Architecture distortion. (**a**) Stellate; (**b**) Circular; (**c**) Powdery; (**d**) Crushed stone-like; (**e**) Casting; (**f**) Architecture distortion.

**Table 1 cancers-12-01855-t001:** Simulated results with respect to proportional interval cancer incidence for breast cancer varying with inter-screening interval and sensitivity.

Inter-Screening	Sensitivity (%)	(IE Ratio × 100) %			
		Stellate	Circular	Powdery & Crushed Stone	Size ≥ 15 mm
**(a) 1977–1985**					
1	40	24	34	29	36
2		42	54	48	56
3		54	66	60	68
1	50	21	30	26	32
2		37	49	43	51
3		49	61	55	63
1	60	19	27	23	29
2		33	44	39	46
3		43	55	50	58
1	80	13	20	17	22
2		24	35	29	37
3		32	45	39	47
1	100	8	14	11	15
2		15	25	20	27
3		21	34	28	37
1	Estimated sensitivity	12	15	12	26
2		22	27	22	42
3		31	37	30	53
**(b) 1996–2010**					
1	40	31	38	29	45
2		51	59	48	65
3		63	70	61	76
1	50	28	35	26	41
2		46	54	43	61
3		58	65	55	71
1	60	25	31	23	37
2		41	49	39	56
3		52	60	50	67
1	80	19	24	17	29
2		32	39	29	46
3		42	50	39	57
1	100	12	17	11	21
2		22	30	20	37
3		31	40	28	48
1	Estimated sensitivity	13	18	11	23
2		23	32	21	39
3		32	43	29	50

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
