# Peer review of "Precision Science on Incidence and Progression of Early-Detected Small Breast Invasive Cancers by Mammographic Features"

_cancers, 2020, doi:10.3390/cancers12071855_

Round 1

Reviewer 1 Report

Chang et al. aimed to estimate the expected incidence (E) in the absence of screening in order to quantify the I/E ratio defined as the observed interval cancer (I) as a percentage of the expected (E) given inter-screening interval and the detectability of mammography (sensitivity). Authors studied two cohorts of patients in whom breast cancer screening: the first period in the era 1977-1985 and the second period in the era 1996-2010. They compared frequencies and percentage of breast cancer cases by different mammographic features in two periods. During the first period only one view mammography was used, while during the second period a two-view mammography was used. Furthermore, different radiological diagnostic tools in both periods probably influenced sensitivity. Unfortunately, they did not compare modern histopathological types as well as molecular subtypes of tumors with different mammographic features. In the era of modern screening programs also other diagnostic techniques are used (for example digital mammography, ultrasound, ultrasound guided core biopsy, vacuum assisted breast biopsy), so their results and conclusions are questionable. My view is that the manuscript is not suitable for the publication in the Cancers.

Reviewer 2 Report

The authors perf a very informative study that could have implications on deciding screening itnervals for breast cancer. While the discussion and conclusions are very clearly presented, the results section is let down by slightly chaotic data presentation. It is extremely hard to follow the text and references to the various numbers and percentages from Table1 and Table 2. Perhaps the authors could revise the text to make the results clearer, and find an alternative method of graphical display of the results.

Reviewer 3 Report

The paper contains an accurate analysis about the incidence of the inter-screening interval on the effectiveness of mammographic screening. The analysis was conducted only on the acinar adenocarcinoma focusing on the stellate type, the circular type, the powdery type and the crushed stone-like type. Nevertheless, it is careful and well done.

Line 138 “…to those for the larger breast tumors larger than 15 mm.” for clarity should be replaced by “… to those for the largest tumors (larger than 15 mm).”

Lines 233-236 should be replaced by “… in the absence of screening. This is the denominator of … (I/E) ratio. This parameter is the main indicator for the evaluation…”

Considerations Inter-screening intervals are strongly dependent by the sensitivity. The sensitivity, in turn, is strongly correlated to the possible adoption of computer-aided systems. It would be useful to clarify if the estimated sensitivity for the second period (1996-2010) is consistent with the sensitivity reachable by the adoption of CAD systems. Nowadays, the same analysis would be more interesting by considering the adoption of new mammographic techniques such as Contrast Enhanced Spectral Mammography (see FANIZZI, Annarita, et al. Fully automated support system for diagnosis of breast cancer in contrast-enhanced spectral mammography images. Journal of clinical medicine, 2019, 8.6: 891.; FANIZZI, Annarita, et al. A machine learning approach on multiscale texture analysis for breast microcalcification diagnosis. BMC bioinformatics, 2020, 21.2: 1-11.)

Anyway, the analysis was conducted on two periods cohort study and provide guidance on mammographic inter-screening intervals

Round 2

Reviewer 1 Report

The revised version of the manuscript also has the same shortcomings as the first version. The authors compared sensitivity of screening mammography in two periods when they used different methods and time intervals of mammography. Major drawback is that authors did not compare modern histopathological types and molecular subtypes of tumors with different mammographic features. So their results and conclusions are questionable in the era of modern screening with digital mammography and other modern diagnostic techniques. My view is that the manuscript is not suitable for the publication in the Cancers.

Round 3

Reviewer 1 Report

The revised version of the manuscript also has the same shortcomings as the first two versions. The authors compared sensitivity of screening mammography in two periods when they used different methods and time intervals of mammography. Major drawback is that authors did not compare modern histopathological types and molecular subtypes of tumors with different mammographic features. So their results and conclusions are questionable in the era of modern screening with digital mammography and other modern diagnostic techniques. My view is that the manuscript is not suitable for the publication in the Cancers.